# Prognostic Impact of Pulmonary Metastasectomy in Bone Sarcoma Patients: A Retrospective, Single-Centre Study

**DOI:** 10.3390/cancers15061733

**Published:** 2023-03-13

**Authors:** Maria Anna Smolle, Angelika Kogler, Dimosthenis Andreou, Susanne Scheipl, Marko Bergovec, Christoph Castellani, Holger Till, Martin Benesch, Florian Posch, Joanna Szkandera, Freyja-Maria Smolle-Jüttner, Andreas Leithner

**Affiliations:** 1Department of Orthopaedics and Trauma, Medical University of Graz, 8036 Graz, Austria; 2Department of Paediatric and Adolescent Surgery, Medical University of Graz, 8036 Graz, Austria; 3Division of Paediatric Haematology/Oncology, Department of Paediatrics and Adolescent Medicine, Medical University of Graz, 8036 Graz, Austria; 4Division of Haematology, Department of Medicine, Medical University of Graz, 8036 Graz, Austria; 5Division of Oncology, Department of Medicine, Medical University of Graz, 8036 Graz, Austria; 6Division of Thoracic and Hyperbaric Surgery, Department of Surgery, Medical University of Graz, 8036 Graz, Austria

**Keywords:** bone sarcoma, metastasis, metastasectomy, survival

## Abstract

**Simple Summary:**

In this retrospective study, the influence of lung metastasectomy on post-metastasis survival (PMS) in bone sarcoma patients was analyzed. Inverse probability of treatment weight (IPTW) was calculated based on a propensity score whether patients underwent metastasectomy or not, in order to account for potential selection bias. We included 47 patients, 39 of whom had undergone metastasectomy. The procedure was more likely to have been performed in singular and unilateral lesions, young patients, and those with a prolonged interval from primary diagnosis to metastasis development. Both in the naive and in the IPTW-adjusted Cox-regression analysis, metastasectomy was associated with a significantly better PMS. Thus, this treatment option should be considered in bone sarcoma patients with pulmonary metastases in order to improve outcomes.

**Abstract:**

This retrospective study aimed at analyzing the impact of metastasectomy on post-metastasis survival (PMS) in bone sarcoma patients with lung metastases. Altogether, 47 bone sarcoma patients (24 males, median age at diagnosis of lung metastases: 21.8 (IQR: 15.6–47.3) years) with primary (n = 8) or secondary (n = 39) lung metastases treated at a single university hospital were retrospectively included. Based on a propensity score, inverse probability of treatment weight (IPTW) was calculated to account for selection bias whether patients had undergone metastasectomy or not. The most common underlying histology was osteosarcoma (n = 37; 78.7%). Metastasectomy was performed in 39 patients (83.0%). Younger patients (*p* = 0.025) with singular (*p* = 0.043) and unilateral lesions (*p* = 0.024), as well as those with an interval ≥ 9 months from primary diagnosis to development of lung metastases (*p* = 0.024) were more likely to undergo metastasectomy. Weighted 1- and 3-year PMS after metastasectomy was 80.8% and 58.3%, compared to 88.5% and 9.1% for patients who did not undergo metastasectomy. Naive Cox-regression analysis demonstrated a significantly prolonged PMS for patients with metastasectomy (HR: 0.142; 95%CI: 0.045–0.450; *p* = 0.001), which was confirmed after IPTW-weighting (HR: 0.279; 95%CI: 0.118–0.662; *p* = 0.004), irrespective of age, time to metastasis, and the number of lesions. In conclusion, metastasectomy should be considered in bone sarcoma patients with lung metastases, after carefully considering the individual risks, to possibly improve PMS.

## 1. Introduction

Metastatic disease is a common cause of death in patients with bone sarcoma [1]. Synchronous metastases are present in 15–20% of osteosarcomas [2,3,4] and up to 25% of Ewing sarcomas [5,6,7], while 20–50% of osteosarcoma and Ewing sarcoma patients will develop secondary metastases, most commonly affecting the lungs [8,9,10]. Apart from chemotherapy (CTX) and radiotherapy (RTX), resection of metastases in general, and lung metastases in particular, has become an important therapeutic option in bone sarcoma patients [11,12], in addition to other local treatment options [13]. Decision in favour of metastasectomy is usually reached after a multidisciplinary treatment team has carefully reviewed individual prognostic factors such as number of metastases, metastasis-free interval, and patients’ general condition [11,12].

With evidence in the literature on the potential effect of pulmonary metastasectomy being based on retrospective studies [12,14,15,16,17,18,19], parameters influencing the decision whether patients undergo metastasectomy or not can only in part have been taken into consideration. On the other hand, prospective studies in bone sarcoma patients with metastatic disease randomising between local metastasectomy versus systemic treatment are difficult to perform due to ethical concerns.

Therefore, this retrospective study aimed at analysing the effect of pulmonary metastasectomy in bone sarcoma patients by applying propensity score and inverse probability of treatment weighting (IPTW) to account for underlying treatment selection bias.

## 2. Materials and Methods

Between 2001 and 2019, 296 consecutive patients with bone sarcoma were surgically treated at a single university hospital. Forty-nine of them had either primary pulmonary metastases (n = 8) or developed lung metastases during follow-up (n = 39) and were included in this retrospective study. Median patient age at diagnosis of pulmonary metastases was 21.8 years (interquartile range (IQR): 15.6–47.3 years). Twenty-four patients were male (51.1%). Median follow-up from primary tumour diagnosis was 50.0 months (IQR: 25.0–93.0 months).

Medical records, surgical, and histopathological reports were used to obtain clinical, surgical, and tumour-related variables. Information on number and location of pulmonary metastases was obtained from radiology reports of the thoracic staging imaging (either X-ray or CT scan) just before decision making regarding metastasectomy. Date of diagnosis of pulmonary metastasis was defined as the first appearance of suspected pulmonary lesions on staging imaging, if (1) they were already present upon initial staging (i.e., prior to any treatment) (2) the suspected lesions progressed with time and/or (3) histological work-up of resected lesions revealed metastases from the underlying bone sarcoma. In patients not undergoing metastasectomy, diagnosis of secondary metastases was confirmed upon progression of suspected pulmonary nodules on two subsequent chest CT-images. This study was approved by the local institutional review board (EK-number: 32-222 ex 19/20).

### Statistical Methods

All statistical analyses were performed with Stata/SE Version 16.1 for Mac (StataCorp, College Station, TX, USA). Normal distribution of variables was assessed with the Shapiro–Wilk test. Means and medians with corresponding standard deviations (SDs) or interquartile ranges (IQRs) were provided for normally and non-normally distributed variables, respectively. T-tests and Wilcoxon rank-sum tests were performed to assess differences in continuous normally or non-normally distributed variables between two groups. Chi-squared tests were performed to assess differences between two binary or binary and categorical variables. *p*-values of <0.05 were considered statistically significant.

To adjust for imbalances between patients undergoing and not undergoing metastasectomy, a propensity score was generated, as previously described [20,21,22]. IPTW was calculated based on the propensity score, defined as the inverse probability of a patient to receive the treatment they actually received [21]. The following variables were included in the propensity score: gender, age at diagnosis of metastases, primary vs. secondary metastases, CTX no/yes, RTX no/yes, bilateral lung metastases no/yes, histology (osteosarcoma vs. Ewing sarcoma vs. others), and pulmonary metastasis-free interval (cut at median of 9 months, i.e., <9 months vs. ≥9 months; calculated from date of biopsy). To assess the extent of balancing following IPTW, standardised mean differences (SMDs) were calculated, with SMDs of variables ≤ 0.3 regarded as sufficiently balanced [20]. After IPTW weighting, univariate and multivariate Cox-regression models were constructed to identify prognostic factors for post-metastasis survival (PMS), which was defined as the time from diagnosis of metastatic disease to last follow-up or death. A *p*-value of <0.05 was considered statistically significant.

## 3. Results

The majority of patients in this study had osteosarcoma (n = 37; 78.7%), followed by Ewing sarcoma (n = 5; 10.7%) and chondrosarcoma (n = 4; 8.5%). One patient had intraosseous leiomyosarcoma (n = 1; 2.1%; Table 1).

Chemotherapy was administered according to COSS (n = 16), EURAMOS-1 (n = 11) and EURO-BOSS (n = 7) protocols in osteosarcoma patients, as well as Ewing 2008 (n = 4) and Euro-Ewing (n = 1) protocols in Ewing sarcoma patients. In one osteosarcoma patient, exact protocol was unknown. Seven patients (five with chondrosarcoma, two with osteosarcoma) did not receive chemotherapy for the primary tumour. Most tumours were localised in the lower extremity (n = 33; 70.2%), followed by pelvis in six and upper limb in five cases, as well as trunk in three patients. In 31 patients with osteosarcoma and 4 patients with Ewing sarcoma, information on necrosis rate of primary tumours following neoadjuvant treatment was available; whilst 12 sarcomas were composed of ≤ 10% vital tumour tissue, more than 10% vital tumour tissue was found in 23.

At final follow-up, 9 patients were alive without disease (19.2%), 16 patients were alive with disease (34.0%), 21 patients had died of their disease (44.7%), and 1 patient had died due to other causes (2.1%). For the entire cohort, one, 3- and 5-year PMS was 87.1%, 55.6%, and 41.1%. Following diagnosis of first pulmonary metastasis, 33 patients (70.2%) developed further metastases, involving the lungs in 24 cases, internal organs in 4, bone in 2, and brain, skin, and soft tissues in 1 case each.

Thirty-nine of the forty-seven patients underwent pulmonary metastasectomy (83.0%), most commonly unilateral wedge resection via thoracotomy (n = 12; 30.8%), and bilateral metastasectomy via sternotomy (n = 6; 15.4%; Appendix A). Reasons not to perform metastasectomy in the remaining eight patients were best supportive care due to progressive disease upon CTX (n = 4), advanced patient age or co-morbidities (n = 2), and additional metastases in the intestine (n = 1) or pleural carcinosis (n = 1) rendering complete surgical remission impossible. Overall, 30 patients were administered CTX for their metastases, including 26 patients with, and 4 without metastasectomy. In 3 patients, metastases were irradiated, including two with metastasectomy, and 1 without.

### 3.1. Difference between Patients Undergoing and Not Undergoing Metastasectomy

With a median age of 19.3 years (IQR: 15.6–40.5 years), patients undergoing resection of pulmonary metastases were 22.5 years younger than patients not undergoing metastasectomy (41.8 years (IQR: 22.0–65.4 years); *p* = 0.025). There was a longer interval between initial tumour diagnosis and development of lung metastases in patients who underwent metastasectomy (12.0 months vs. 2.5 months for patients without metastasectomy; *p* = 0.023). Of note, patients with and without primary pulmonary metastases were equally likely to undergo metastasectomy (*p* = 0.091), whilst patients with singular metastases were more likely to be treated by metastasectomy (*p* = 0.043; Table 1). Patients with bilateral pulmonary metastases were less likely to undergo metastasectomy, despite the fact that 17 patients, who finally underwent resection of pulmonary metastases, had lesions in both lungs (*p* = 0.024). Patients receiving CTX appeared more likely to undergo metastasectomy (*p* = 0.049). There were no significant differences in histological subtype between patients with and without metastasectomy (*p* = 0.329; Table 1).

IPTW weighting of the data reduced imbalances between variables gender, presence of primary lung metastases, histology, CTX, and RTX to a sufficient amount (all SMDs ≤ 0.3). However, residual imbalances persisted for variables time to pulmonary metastases, age at diagnosis of pulmonary metastases, bilateral lung metastases, and multiple metastases (SMDs > 0.3; Table 1).

### 3.2. Difference in Post-Metastasis Survival

In the naïve univariate Cox-regression analysis, patients undergoing metastasectomy had a significantly better PMS than patients not undergoing metastasectomy (hazard ratio [HR]: 0.142; 95% confidence interval [CI]: 0.045–0.450; *p* = 0.001). This effect was likewise present when excluding the eight patients with primary pulmonary metastases (HR: 0.084; 95%CI: 0.015–0.464; *p* = 0.005). After weighting the data for the IPTW, the positive influence of metastasectomy on PMS—albeit slightly less pronounced—prevailed (HR: 0.237; 95%CI: 0.090–0.623; *p* = 0.003; Table 2).

Weighted one-, three- and five-year PMS was 80.8%, 58.3%, and 44.0% for patients with metastasectomy, compared to 88.5% and 9.1% at one and 3 years for patients without metastasectomy (last patient in this cohort lost to follow-up after 29 months; Figure 1).

Patients with multiple metastases had a significantly worse PMS than patients with solitary lung metastases (HR: 2.858; 95%CI: 1.173–6.967; *p* = 0.021). The same was the case for a metastasis-free interval of <9 months compared to an interval of ≥9 months (HR: 0.255; 95%CI: 0.114–0.573; *p* = 0.001). There was no significant difference in PMS between patients with underlying osteosarcoma, Ewing sarcoma, and other histologies (Table 2).

Due to the remaining imbalances after IPTW weighting, reflected by insufficient reduction in SMDs for some variables, multivariate analyses taking into consideration time to pulmonary metastases, patient age, and singular vs. multiple metastases, were performed. In this multivariate Cox-regression model, the significant positive influence of metastasectomy towards improved PMS prevailed (HR: 0.279; 95%CI: 0.118–0.662; *p* = 0.004; Table 2), irrespective of patient age (*p* < 0.001), time to pulmonary metastasis (*p* = 0.003), and presence of multiple metastases (*p* = 0.798).

### 3.3. Long-Term Post-Metastasis Survival

Twelve patients had a PMS of at least 5 years (median 98.0 months (IQR: 81.5–117.0 months)). All of them had undergone pulmonary metastasectomy (Table 3). Seven of the twelve patients had more than one pulmonary lesion at time of diagnosis of metastatic disease. Two patients presented with primary lung metastases, whilst the remaining ten developed secondary pulmonary lesions. Nine patients had been diagnosed with osteosarcoma, two with chondrosarcoma, and one with Ewing sarcoma. Six patients developed further metastases following initial metastasectomy (Table 3), and four of these subsequently underwent another metastasectomy. One patient with osteosarcoma died of disease at 109 months post metastasectomy, four were alive with disease, and seven were alive without disease (Table 3).

## 4. Discussion

The current retrospective single centre study demonstrated that pulmonary metastasectomy is associated with a prolonged post-metastasis survival in bone sarcoma patients, irrespective of favourable prognostic factors prevailing at baseline in patients undergoing metastasectomy. In particular, younger patients with a prolonged metastasis-free interval seemed to benefit from pulmonary metastasectomy.

Any therapy requires careful weighting of risks and benefits in each individual patient, with more invasive treatments eventually being associated with a higher morbidity. In comparison to patients with unfavourable prognosis and in poor general condition, healthy patients with a good prognosis are naturally more likely to undergo invasive therapies with predictable side-effects. In line with this, we discovered significant differences at baseline depending on whether patients underwent metastasectomy, a treatment associated with anaesthesiologic and surgical risks. For example, patients with bilateral lung involvement and multiple metastases were less likely to undergo metastasectomy, as were older patients and those with short metastasis-free interval. All these variables are known for their prognostic impact, with advanced age, multiple nodules, and short interval from diagnosis to development of metastases being associated with poor survival in bone sarcoma patients [16,17,23]. Consequently, a considerable treatment selection bias has to be taken into account when interpreting the effect of pulmonary metastasectomy. We therefore applied IPTW to account for differences at baseline between patients with and without pulmonary metastasectomy.

In our study, IPTW-weighted 5-year PMS probability for the 39 patients with pulmonary metastasectomy was 44.8%. This rate is higher than the 22.6%, 31%, and 38% reported by Harting et al., [14], Chen et al. [15], and Buddingh et al. [24], but comparable to the 41.0% and 46.2% discovered by Liu et al. [25] and Kim et al. [17]. In the study by Liu et al., data had likewise been weighted for an IPTW to reduce baseline confounding regarding metastasectomy assignment [25]. Of note, all the studies mentioned above reported exclusively on osteosarcoma patients with lung metastases, whereas our study also included a small proportion of patients with other bone sarcomas.

As residual imbalances between baseline variables prevailed after IPTW adjustment, we performed multivariate Cox-regression analyses including factors significantly associated with PMS in the univariate analysis. In the resulting model, pulmonary metastasectomy was significantly associated with a prolonged PMS, irrespective of patient age, presence of multiple metastases, and time to development of pulmonary lesions. These results are contrary to the ones made by Liu et al., who reported no significant impact of metastasis-free interval and patient age on PMS [25]. However, the herein observed significant impact of age on PMS may be explained by the fact that we likewise included paediatric and adult patients–reflected by a median age of 21.8 years at diagnosis, whilst Liu et al. analysed paediatric patients only, with a median age of 17 years [25]. Furthermore, irrespective of singular vs. multiple pulmonary lesions, bone sarcoma patients in our cohort seemed to benefit from metastasectomy, corroborating the findings by Liu et al. [25]. Related to this, 12 of the 47 patients (all in the metastasectomy group, eight with osteosarcoma) in our cohort survived at least 5 years following diagnosis of pulmonary metastases. At a median PMS of 8.2 years, seven of them were alive without disease, four were alive with disease, and one had died of disease. This strengthens the assumption that long-term survival with potential curative intent in bone sarcoma patients with pulmonary metastases is possible.

Due to the limited number of patients available, the authors analysed patients with different bone sarcomas jointly, albeit the majority of patients had osteosarcoma (76.5%). As no significant difference between patients with and without metastasectomy depending on the underlying histological subtype was present, and that histology was likewise included in the propensity score for later IPTW adjustment, it can be assumed that the observed protective effects of pulmonary metastasectomy are likewise applicable to either histological subtype. However, the limited number of Ewing sarcoma did not allow for subgroup analysis of different histological subtypes. Nevertheless, similar results have been reported by Letourneau et al. and Tronc et al., with pulmonary metastasectomy being associated with a significant survival improvement in Ewing sarcoma patients [18,26]. Yet again, the number of Ewing sarcoma patients in either cohort was small, raising the need for further studies focusing on Ewing sarcoma in particular.

A major limitation of the current study is its retrospective design with non-random assignment of patients to pulmonary metastasectomy. Thus, significant imbalances in baseline variables were present, with favourable prognostic factors prevailing in patients undergoing metastasectomy. These imbalances could be—to a great extent—adjusted for by applying propensity score and IPTW. Moreover, the impact of different surgical approaches on PMS could not be analysed due to the low number of patients in each surgical group. Another limitation of the study is the small number of patients necessitating larger scaled studies are warranted to further elucidate our findings.

## 5. Conclusions

In conclusion, pulmonary metastasectomy in bone sarcoma patients is associated with a survival benefit, persisting even after adjustment for favourable prognostic factors prevailing in patients undergoing surgery for metastatic nodules. Although these benefits have to be weighted carefully against the risks associated with the procedure, and young bone sarcoma patients with a prolonged metastasis-free interval in particular should be offered resection of pulmonary metastases with a potential curative intent, whilst the number of metastatic lesions may not be a decisive factor for metastasectomy.

## Figures and Tables

**Figure 1 cancers-15-01733-f001:**
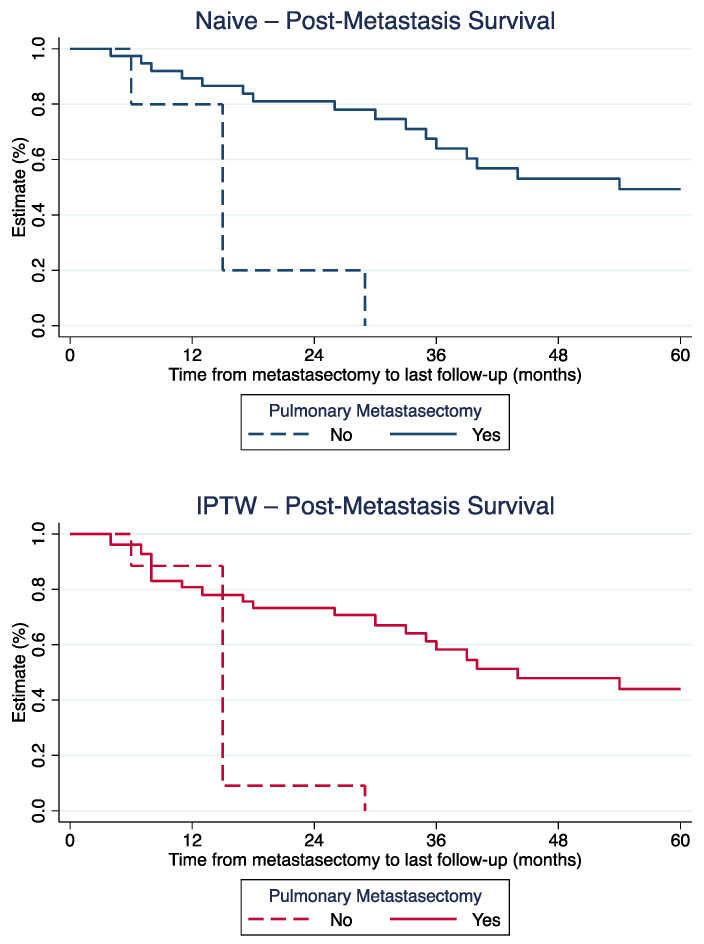
Post-metastasis survival for patients with and without metastasectomy, before (**top**) and after (**bottom**) IPTW-adjustment.

**Table 1 cancers-15-01733-t001:** Differences at baseline between patients with and without metastasectomy.

		Total (%)	Metastasectomy		*p*-Value	SMD	SMD after IPTW
			No (n = 8)	Yes (n = 39)			
Gender	Male	24 (51.1)	3	21	0.400	0.32	0.02
	Female	23 (48.9)	5	18			
Age at diagnosis of metastases (in years; median + IQR)		21.8 [15.6–47.3]	41.8 [22.0–65.4]	19.3 [15.6–40.5]	0.025 *	0.84	0.34
Primary lung metastases	No	39 (83.0)	5	33	0.091	0.56	0.17
	Yes	8 (17.0)	3	5			
Time to lung metastases * (in months; median + IQR)		9 [3–28]	2.5 [0–8.5]	12 [6–37]	0.023 *	N/A	N/A
Time to lung metastases **	<9 months	24 (51.1)	7	17	0.024 *	1.01	0.52
	≥9 months	23 (48.9)	1	22			
Histology	Osteosarcoma	37 (78.8)	5	32	0.329	0.43	0.11
	Ewing sarcoma	5 (10.6)	1	4		0.07	0.15
	Others	5 (10.6)	2	3		0.46	0.02
RTX ***	No	43 (91.5)	7	36	0.657	0.15	0.08
	Yes	4 (8.5)	1	3			
CTX	No	7 (14.9)	3	4	0.049 *	0.64	0.06
	Yes	40 (85.1)	5	35			
Multiple pulmonary metastases	No	14 (29.8)	0	14	0.043 *	1.04	0.91
	Yes	33 (70.2)	8	25			
Bilateral involvement	No	23 (48.9)	1	22	0.024 *	1.01	0.99
	Yes	24 (52.1)	7	17			

* significant result; Legend: CTX—chemotherapy; IQR—interquartile range; RTX—radiotherapy; SMD—standardised mean difference. *p*-values obtained with Wilcoxon rank-sum tests for continuous variables and chi-squared tests for binary variables. * not used for propensity score; ** used for propensity score; *** for primary tumour.

**Table 2 cancers-15-01733-t002:** Univariate and multivariate IPTW-weighted Cox-regression models for post metastasis survival.

		HR	95% Confidence Interval		*p*-Value
			Lower	Upper	
**Univariate**					
Metastasectomy	No	1			0.003 **
	Yes	0.237	0.090	0.623	
Gender	Male	1			0.360
	Female	1.499	0.630	3.567	
Age at diagnosis of metastasis (in years)		1.035	1.014	1.056	0.001 **
Time to lung metastases	<9 months	1			0.001 **
	≥9 months	0.255	0.114	0.573	
Multiple metastases	No	1			0.021 **
	Yes	2.858	1.173	6.967	
Histology	Osteosarcoma	1			
	Ewing sarcoma	0.522	0.122	2.224	0.380
	Others	2.075	0.609	7.073	0.243
RTX *	No	1			0.557
	Yes	0.658	0.163	2.661	
CTX	No	1			0.068
	Yes	0.347	0.111	1.083	
**Multivariate**					
Metastasectomy	No	1			0.004 **
	Yes	0.279	0.118	0.662	
Age at diagnosis of metastasis (in years)		1.054	1.034	1.0.73	<0.001 **
Time to lung metastases	<9 months	1			0.003 **
	≥9 months	0.214	0.077	0.595	
Multiple metastases	No	1			0.798
	Yes	0.860	0.273	2.714	

* for primary tumour; ** significant result.

**Table 3 cancers-15-01733-t003:** Long-term survivors following pulmonary metastasectomy.

1	Age at Lung Metastasis Diagnosis (in Years)	Diagnosis	Time from Primary Diagnosis to Metastasis (in Months)	Number of Metastases	Bilateral Involvement	Type of Metastasectomy	PMS (in Months)	2nd Metastasis	Outcome
M	33	Chondrosarcoma, left distal femur	60	≥5	1	Thoracotomy, bilateral wedge resection	75	Yes, lungs	AWD
F	43	Chondrosarcoma, left hemithorax	46	≥5	0	Thoracotomy, unilateral wedge resection	89	Yes, lungs	AWD
M	15	Ewing sarcoma, right ulna	8	1	0	Thoracotomy, unilateral wedge resection	123	No	NED
M	15	Osteosarcoma, left distal femur	0	2	0	Sternotomy, unilateral metastasectomy	88	No	NED
F	24	High-grade osteosarcoma, left calcaneus	37	1	0	Video-assisted thoracotomy, unilateral wedge resection	80	No	NED
M	15	Osteosarcoma, left distal femur	67	1	0	Thoracotomy, unilateral wedge resection	107	Yes, lungs, thorax	NED
M	11	Osteosarcoma, left distal femur	23	1	0	Sternotomy, lobectomy	135	Yes, brain	AWD
M	17	Osteosarcoma, left distal femur	26	≥5	1	Segmental resection	171	No	NED
M	13	Osteosarcoma (chondroblastic), left distal femur	6	≥5	1	Sternotomy, bilateral metastasectomy	109	Yes, lungs	DOD
M	16	High-grade osteosarcoma, right femur diaphysis	18	1	0	Thoracotomy, unilateral wedge resection	66	No	NED
M	19	Osteosarcoma (osteoblastic), left distal tibia	0	4	1	Sternotomy, bilateral metastasectomy *	111	No	NED
M	21	Osteosarcoma, right distal femur	38	2	0	Sternotomy, unilateral metastasectomy	83	Yes, lungs	AWD

Legend: AWD—alive with disease; DOD—dead of disease; F—female; M—male; NED—no evidence of disease; PMS—post metastasis survival. * no remaining malignant tumour tissue found upon histological work-up.

## Data Availability

Original data can be obtained upon reasonable request from the corresponding author.

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
