# Peer review of "Prognostic Impact of Pulmonary Metastasectomy in Bone Sarcoma Patients: A Retrospective, Single-Centre Study"

_cancers, 2023, doi:10.3390/cancers15061733_

Round 1

Reviewer 1 Report

Thank you for allowing me to review this study examining the impact of metastasectomy in patients with a bone sarcoma. This is an interesting topic to the readers. I think this contains data that is applicable to everyone in the sarcoma field

Specific questions:

- When do you perform metastasectomy? Is it after all chemo? Before you start? Was there a different approach based on histology?

-What was the tumor necrosis in the primary tumor and the metastatic disease? Do you have that information?

- Did you use any other treatment for the mets such as radiation? In Table 1 was the radiation to the primary tumor or to the mets?

-What were your chemos?

- I may have missed it in the paper. How did patient survival who didn't present with mets compare to patients who had a pulmonary met and underwent metastasectomy compare?

Author Response

Thank you for allowing me to review this study examining the impact of metastasectomy in patients with a bone sarcoma. This is an interesting topic to the readers. I think this contains data that is applicable to everyone in the sarcoma field.

Author Response: The authors would like to thank the reviewer for their time and all valuable improvement suggestions made. They have been incorporated at best possible in the updated version of the manuscript.

Specific questions:

- When do you perform metastasectomy? Is it after all chemo? Before you start? Was there a different approach based on histology?

Author Response: Thank you for this important comment. At our institution, decision on metastasectomy is made interdisciplinary, taking into consideration extent of disease, histology, patients’ general condition, and expected prognosis. In tumours of low chemo-sensitivity (e.g. chondrosarcoma), metastasectomy is the preferred primary treatment option. On the other hand, in Ewing sarcoma and osteosarcoma, metastasectomy is usually combined with systemic treatment, with chemotherapy administered prior to surgical removal of metastases.

-What was the tumor necrosis in the primary tumor and the metastatic disease? Do you have that information?

Author Response: Thank you for this valuable comment. Information on necrosis rate of osteosarcomas and Ewing sarcomas following neoadjuvant treatment could be ascertained from pathology reports in 31/37 and 5/5 cases. In detail, 12 tumours contained less than 10% vital tumour tissue, whilst in 23 tumours, more than 10% vital tumour tissue following neoadjuvant therapy was present. This information has been added to the results section of the manuscript. For resected metastases, no such information was available, though.

(Results, lines 129 – 132)

- Did you use any other treatment for the mets such as radiation? In Table 1 was the radiation to the primary tumor or to the mets?

Author Response: Radiotherapy in Table 1 refers to irradiation applied to the primary tumour. This is now clarified in Table 1 and 2 via a footnote. Three patients underwent radiotherapy for their metastases (two in addition to pulmonary metastasectomy, and one as only palliative treatment). Furthermore, 30 patients received chemotherapy for their metastases, including 4 patients without metastasectomy, and 26 with metastasectomy. This information has likewise been added to the results section of the manuscript.

(Results, lines 143 – 147; Table 1; Table 2).

-What were your chemos?

Author Response: Chemotherapy protocols for primary tumours included COSS (n=16), EURAMOS-1 (n=11) and EURO-BOSS (n=7) for osteosarcoma (in one osteosarcoma patient, chemotherapy protocol was unknown), as well as Ewing 2008 (n=4) and Euro-Ewing (n=1) for Ewing sarcoma. This information has been added to the results section of the manuscript.

(Results, lines 123 – 127)

- I may have missed it in the paper. How did patient survival who didn't present with mets compare to patients who had a pulmonary met and underwent metastasectomy compare?

Author Response: The current study only included patients who developed pulmonary metastases at some time point. However, any bone sarcoma patient without primary/secondary metastases was excluded, as the purpose of the study was to assess the effect of pulmonary metastasectomy in patients with metastases.

Reviewer 2 Report

While a prospective randomized trial would be the ideal study design to answer the question of whether metastasectomy portends a survival benefit, the authors address early in their manuscript the ethical challenges involved in that. They make their best attempt at correcting for confounding/treatment selection bias that exists in their retrospective data set. They also carefully word their Discussion and Conclusion to explain that an association (but not a causal relationship) was what they detected between survival and metastasectomy. With their acknowledgement of their study's weaknesses, I believe this is worthy of publication.

Author Response

While a prospective randomized trial would be the ideal study design to answer the question of whether metastasectomy portends a survival benefit, the authors address early in their manuscript the ethical challenges involved in that. They make their best attempt at correcting for confounding/treatment selection bias that exists in their retrospective data set. They also carefully word their Discussion and Conclusion to explain that an association (but not a causal relationship) was what they detected between survival and metastasectomy. With their acknowledgement of their study's weaknesses, I believe this is worthy of publication.

Author Response: The authors would like to thank the reviewer for their time and the overall positive appraisal of the current manuscript. The authors are aware of the fact that retrospective studies harbor several limitations in comparison to any prospective (randomized) study that can only in part be compensated for upon thorough statistical analysis. Yet, by applying propensity score measures, the treatment selection bias towards metastasectomy in this retrospective setting was taken into consideration at best possible.

Based on comments made by other reviewers, some minor changes were made to the manuscript.

Reviewer 3 Report

Thank you for inviting the review for the study about pulmonary metastasectomy in bone sarcoma patients. The authors demonstrated the improvement of post-metastasis survival by metastasectomy in bone sarcoma. The most important results of this study was shown in Table2, metastasectomy, age, time to lung metastasis and multiple metastasis were independent prognostic factors.

I thought there several points for the study.

1.     In Figure 1, type of pulmonary metasectomy was demonstrated. I thought this Figure was not needed for this study. From this result, the discussion about the results should be given. If not important, change the figure to supplement or combin with Table 1.

2.     In Table2, about post-metastasis survival, chemotherapy and radiotherapy should be included multivariate analysis. Because, in osteosarcoma and Ewing sarcoma that they were majority of this cohort, those treatment were generally effective (CTX for OS and Ewing, RTX for Ewing).

Author Response

Thank you for inviting the review for the study about pulmonary metastasectomy in bone sarcoma patients. The authors demonstrated the improvement of post-metastasis survival by metastasectomy in bone sarcoma. The most important results of this study was shown in Table2, metastasectomy, age, time to lung metastasis and multiple metastasis were independent prognostic factors.

Author Response: The authors would like to thank the reviewer for their valuable time and all improvement suggestions made. They have been thoroughly included in the revised version of the manuscript.

I thought there several points for the study.

  1. In Figure 1, type of pulmonary metasectomy was demonstrated. I thought this Figure was not needed for this study. From this result, the discussion about the results should be given. If not important, change the figure to supplement or combin with Table 1.

Author Response: As suggested by the reviewer, Figure 1 has been moved to supplementary material as not containing the most integral information of the current study.

  1. In Table2, about post-metastasis survival, chemotherapy and radiotherapy should be included multivariate analysis. Because, in osteosarcoma and Ewing sarcoma that they were majority of this cohort, those treatment were generally effective (CTX for OS and Ewing, RTX for Ewing).

Author Response: Thank you for this important comment. Indeed, the results upon univariate analysis of the effect of chemo- and radiotherapy on post-metastasis survival in bone sarcoma patients were not presented in the initial version of the manuscript. These have now been added to the manuscript in Table 2. However, as both variables turned out as non-significant, they were not considered in the multivariate model.

(Results, Table 2)

Round 2

Reviewer 1 Report

Thank you for your corrections. Would suggest adding these references since they go to your comments about pulmonary nodules in patients with bone sarcomas

PMID: 32956141

  • PMID: 33717939